# Pharmacists’ Role in Older Adults’ Medication Regimen Complexity: A Systematic Review

**DOI:** 10.3390/ijerph18168824

**Published:** 2021-08-21

**Authors:** Catharine Falch, Gilberto Alves

**Affiliations:** CICS-UBI-Health Sciences Research Centre, University of Beira Interior, Av. Infante D. Henrique, 6200-506 Covilhã, Portugal; cathyfalch@gmail.com

**Keywords:** older adults, medication regimen complexity, pharmacist, systematic review

## Abstract

Medication regimen complexity (MRC) may influence health outcomes, such as hospitalisation, hospital readmission and medication adherence. Pharmacists have been referred to as health professionals with the opportunity to act on MRC reduction. This study aimed to investigate pharmacists’ role in studies about older adults’ medication regimen complexity. A literature search was performed in PubMed, Web of Science and the Cochrane Library—CENTRAL—up to October 2019. Out of 653 potentially relevant studies, 17 articles met the inclusion criteria for this review. Most studies used the 65-item medication regimen complexity index (MRCI) to assess medication complexity. Pharmacists’ role was mainly confined to data collection. It seems that pharmacists’ active role in older adults’ medication complexity has not been studied in depth so far. However, the few existing interventional ones suggest that, after previous training, regimen simplification is feasible.

## 1. Introduction

Nowadays the world faces global ageing, often associated with a high prevalence of multimorbidities. As a natural consequence, older age frequently stands out for polypharmacy and complex medication regimens [1,2,3]. When considering medication regimen complexity (MRC), there is so far no clear definition for it [4,5]. However, it has already been shown that the number of drugs is only one of the relevant factors to consider, and that, in addition, dosage form, dosage frequency and administration instructions also have to be considered [6,7,8,9,10,11,12,13,14]. Furthermore, there is also no agreement about the reference instrument for MRC determination [4,5]. Several tools have been used, with the 65-item medication regimen complexity index (MRCI), developed by George et al. [15], being the most common, reliable and validated tool for this purpose, which has already been translated and validated to a few languages [16,17,18] and even standardised for older adults in primary care [19]. It is an open-ended index, with higher total MRCI scores representing more complex medication regimens.

Interest in this subject has grown because numerous studies have associated high MRC with non-adherence [4,20,21], higher hospitalisation rates [20], hospital discharge destination different than home [22] and low overall quality of life [23]. Indeed, medication management may frequently be demanding for the older population, their caregivers and healthcare professionals. In fact, older adults often present reduced manual dexterity in addition to cognitive and sensory impairment that can lead to a higher risk of medication errors and drug-related problems (DRPs) [2,3,6,7,8]. In order to reduce these negative aspects, it seems imperative to attempt medication regimen simplifications in many circumstances. Some research has already investigated its feasibility, with evidence suggesting that complexity can be reduced, and referring to pharmacists as healthcare professionals with a great potential to perform it in routine pharmaceutical dispensing or as part of medication reviews [2,9,11,13,14]. Indeed, pharmacists have a privileged access to the population’s medication, both in community pharmacies and hospital settings, and awareness of this topic is needed, especially in the older population, for whom managing their daily medication may often represent a considerable challenge.

Up to the present date, to the best of our knowledge, there are no systematic reviews available about the role that pharmacists play in the older population’s MRC and the effort made to simplify it. To address this gap, this study aimed to examine and describe pharmacists’ role in studies on older adults’ MRC.

## 2. Materials and Methods

### 2.1. Search Strategy

A systematic literature search was conducted in three databases (PubMed, Web of Science and the Cochrane Library—CENTRAL) from their inceptions to October 2019. The search strategy considered the PICOS elements, representing P—Population, older adults; I—Intervention, pharmacists’ role in MRC; O—Outcomes of interest, any outcome related to MRC; and S—Study design, original peer-reviewed observational or experimental studies. The comprehensive search expression included the combination of keywords related to pharmacists (e.g., pharmacists, pharmaceutical services/care/intervention), older people (e.g., aged, elderly, old age, geriatric, retired, ancient) and medication regimen complexity (e.g., treatment/medicine/drug complexity) [24]. The detailed PubMed search strategy is provided in Appendix A. To ensure literature saturation, references lists from included articles were screened for potential further relevant studies. The PRISMA guideline was used to perform and report items in the present review [25,26].

### 2.2. Study Selection

To be included in this review the article had to meet the following criteria: all study participants had to be aged 60 or over, since, according to the World Health Organisation (WHO), the definition of older person “varies among countries but is often associated with the age of normal retirement” (60 or 65 years) [27];pharmacists’ role in MRC had to be mentioned in the article (for this purpose all pharmacists’ actions were regarded, beginning with simple data collection and ending with pharmacist intervention). Sabater’s et al. [28] definition of pharmacist intervention was considered: “pharmacist intervention is defined as the pharmacist’s activity consisting in a suggested action on the patient treatment and/or an action on the patient oriented towards finding a solution for or preventing a negative clinical outcome of the pharmacotherapy”;medication complexity had to be assessed, and for this purpose all tools were considered;be an original peer-reviewed observational study (i.e., cohort study, cross-sectional study, case study) or an experimental study (i.e., randomised controlled trial, quasi-experimental study);be written in English, Portuguese or German.

Articles were excluded if they:were not performed exclusively on the older population;did not mention any role of pharmacists in MRC, or MRC assessment was not performed;were qualitative studies, reviews, protocols, congress abstracts, editorials, letters, dissertations, theses, feasibility or pilot studies;did not fulfil the language restrictions.

Additionally, no timing or setting restrictions were applied, and only published studies were included.

### 2.3. Data Extraction

Literature search results were uploaded to the Covidence platform, removing duplicates using the “duplicate” function. The remaining duplicates were removed manually.

Two independent reviewers (CF and GA) analysed the studies by screening titles and abstracts to verify potential inclusion criteria correspondence. If an article potentially met the inclusion criteria or provided insufficient information in the abstract to be excluded, the full text was obtained and screened by the same investigators. Any disagreement between reviewers was solved through discussion. Data extraction was performed using a previously created data extraction form (Microsoft Word format) by a single reviewer (CF) and was independently checked afterwards by the second reviewer (GA). Reviewers were not blinded to the authors or journals when screening articles and extracting data.

The following information was collected: authors, year of publication, country, study design, participants’ demographics (no. of participants, age, gender), setting, study aim, medication data (source, type of medication included, instrument to assess MRC), pharmacists’ role and main outcomes.

### 2.4. Quality Assessment

To evaluate the risk of bias and study design, two reviewers (CF and GA) independently used the Effective Public Health Practice Project (EPHPP) quality assessment tool [29]. Any disagreement between reviewers was to be solved by discussion.

The EPHPP quality assessment tool is a generic tool that assesses six domains (a. selection bias, b. study design, c. confounders, d. blinding, e. data collection methods and f. withdrawals and dropouts); it was chosen because of its applicability in a variety of study designs [30].

## 3. Results

A total of 653 potentially relevant studies were identified from the databases. After title and abstract screening in addition to full-text assessment against inclusion criteria, 16 studies remained to be included in this systematic review. One study was identified from the searching reference list and added to the selection, resulting in 17 included studies (Figure 1) [10,12,22,31,32,33,34,35,36,37,38,39,40,41,42,43,44]. 

### 3.1. Study Characteristics 

Characteristics of the included studies in this systematic review are shown in Table 1 and Table 2. The included studies were performed in Australia (*n* = 7) [10,12,22,33,34,42,44], Brazil (*n* = 2) [31,41], Israel (*n* = 1) [39], Spain (*n* = 1) [43] and the United States (*n* = 6) [32,35,36,37,38,40], and were reported between 1990 and 2018. Most articles were written in English, except one that was written in Portuguese [31]. In regard to study design, most studies were cross-sectional (*n* = 7) [31,32,36,37,38,41,43] and cohort studies (*n* = 8) [10,12,22,33,34,39,42,44], but a quasi-experimental (*n* = 1) [40] and a prospective controlled trial (*n* = 1) [35] were also included. The majority of the studies were conducted in health care settings: hospital wards/units/clinics (*n* = 10) [10,12,22,33,34,35,38,39,43,44], primary health care units (*n* = 1) [41], continuing care retirement communities (*n* = 1) [36] and a residential aged care facility (*n* = 1) [42], but data were also obtained by home visits (*n* = 2) [31,37], in churches (*n* = 1) [32] or through a telephone consultation (*n* = 1) [40]. The total number of participants included in the 17 articles was 3652, with sample sizes ranging from 79 to 400 individuals. Participants’ mean age ranged from 71.2 to 86.8 years, with females representing between 39.5 and 79.8% of participants.

### 3.2. Study Quality

Concerning quality assessment, using the EPHPP global rating decision tool, only two studies were rated as being of strong quality [33,34], eight of moderate [10,12,22,34,35,39,40,42] and seven of weak quality [31,32,36,37,38,41,43] (Appendix A). 

### 3.3. Medication Regimen Complexity Assessment

To assess MRC, six studies considered only prescription medication [31,32,33,35,40,42], while four included both prescription and non-prescription medication [22,36,38,44]. Other studies did not mention the prescription/non-prescription status of medication, considering instead long-term, short-term and “when required” medication (*n* = 3) [10,12,34], or only long-term medication (*n* = 1) [39]. Additionally, one study included all medication that subjects take a day, without any other mention [37] and another one included patients’ routine chronic medication [43].

In regard to the instrument used to assess MRC, the majority of studies (*n* = 11) [10,12,22,32,33,39,40,41,42,43,44] applied the 65-item MRCI, which is an open-ended tool, where the final score is the result of the sum of three sections (dosage forms, dosage frequency and additional instructions). Besides this, one study used the medication complexity index (MCI) [31], two studies calculated a complexity score by summing the different dosage intervals, weighted for frequency [35,36], one study calculated the number of times that medications were taken in a 24 h period for each subject [37] and another study did not make reference to the instrument used to determine regimen complexity [34]. Additionally, one study [38] used the patient-level MRCI (pMRCI), which is the sum of three MRCI sub-scores for: prescription disease state medications, prescription for other non-disease medications and over-the-counter (OTC) medications [45,46]. 

### 3.4. Pharmacists’ Role on Older People’s Medication Regimen Complexity and Main Outcomes When Intervention Is Performed

In most studies pharmacists’ role was limited to data collection (*n* = 8) [22,31,33,35,36,39,41,44]. In two studies, pharmacists acted as coders [37,38] and in two other they contributed to data analysis [32,43]. Four studies referred specifically to pharmacists’ actions on MRC [10,12,34,42]: in one, pharmacists only determined regimen simplification potential [10], in two [12,34], simplifications were implemented, and at last, in the fourth study [42], the impact of pharmacists’ residential medication management reviews (RMMRs) on the MRCI were retrospectively analysed. Furthermore, another study reported pharmacists’ intervention on medication- and health-related problems (MHRPs), with MRC being one of the health-related covariates that could be changed [40].

Concerning the main outcomes when pharmacist intervention is performed, Elliot [34] states the proportion of identified and implemented regimen simplifications, and the reasons for non-implementation, as endpoints, while Elliot et al. [12] consider the change in MRCI score between hospital admission and discharge as main outcome measures. Moreover, Pouranayatihosseinabad et al. [42] present their outcomes as MRCI score change (at baseline, after pharmacists’ recommendation and after general practitioners (GPs) acceptance of those recommendations) and number and type of pharmacists’ recommendations and further GPs’ acceptance. Finally, Moczygemba et al. [40] present their study results as clinical (change in MHRPs and medication adherence) and economic outcomes (Part D drug costs). 

Lack of time [12,34], non-acceptance of recommendations by the physicians and patients [12,34,35] as well as medication prescribed by another physician [35] were pointed out as the main reasons for non-implementation of regimen simplifications.

## 4. Discussion

Three recent systematic reviews focused on MRC: Wimmer et al. [20] reviewed the association of clinical outcomes with MRC in older people, Pantuzza et al. [4] investigated the association between MRC and pharmacotherapy adherence and Alves-Conceição et al. [47] identified health outcomes related to MRC measured by MRCI. 

This work is the first systematic review to explore pharmacists’ role in studies on MRC of the older population to the best of our knowledge.

### 4.1. Medication Regimen Complexity Assessment

At first, it is essential to mention the heterogeneity of instruments used to assess MRC. Several instruments were used in the different studies, including the medication complexity index (MCI) [31], which failed to show satisfactory reliability with complex regimens, and did not demonstrate any significant correlation with outcomes such as medication adherence [15,48]. However, most studies already use the MRCI, which already shows good evidence of classifying complexity better than a simple medication count [6], discriminating between regimens with an equal number of medications, resulting in higher complexity scores for regimens with fewer drugs [15] and being a better overall predictor of all-cause mortality [49] and discharge destination [22] than polypharmacy. Additionally, in a few studies, the MRCI has been regarded as beneficial in targeting patients who may benefit from additional services such as domiciliary reviews and medication therapy management (MTM) services [14,15,45]. These strengths of the MRCI over other instruments should be taken into account in future investigations, especially regarding the importance of using a universal tool for MRC determination.

A greater consensus should also be achieved about the type of medication included to determine regimen complexity, which varied from prescription to non-prescription; long-term, short-term and “when required” as well as routine chronic medication. Even concerning the MRCI, there is no uniformity in the medications to be included. Although the instrument was initially developed and validated only for prescribed medications [15], several studies already indicate that prescription and non-prescription medications contribute to regimen complexity and should be considered [14,20,22,44,45,46]. However, even in that case, there is still no harmony in the practical applicability of the instrument: some authors [22,44] use the original MRCI, while Linnebur et al. [38] use the pMRCI. This aspect may be relevant to set high and low complexity scores, which has not yet been achieved despite some research in that area [9].

### 4.2. Measured Outcomes

Regarding the overall measured outcomes, in pharmacy practice research, the Economic, Clinical and Humanistic Outcomes (ECHO) model should be followed, with clinically meaningful outcomes being the most desirable [50,51]. However, in the present review none of the included studies present their results entirely according to this recommendation. Despite that, most of the reported results were related to the type of regimen simplification and its feasibility, reasons for non-implementation, change in the MRCI, the effect of recommendations as well as knowledge and preference of patients, which are endpoints whose relation to better patient outcomes are unknown [51]. Collection and further publication of relevant outcomes should be considered in future research.

### 4.3. Study Setting

In contrast with what was expected, most of the included studies were conducted in hospitals or clinics, but none in community pharmacies. This can reflect different factors: on the one hand there may exist an underreporting of provided pharmaceutical services, while on the other hand, it is also possible that still little attention has been given to this subject, even though several studies already state that the MRCI may be a valuable tool to prioritise patients who could take advantage of medication reviews or drug therapy management services [14,15,45]. At that time, MRC determination tools can be included, side by side with those that identify potentially inappropriate medications (PIMs), such as Beers [52] and STOPP criteria [53], as starting points for medication reduction, which are already an onset for regimen simplification. The frequently polymedicated older population may benefit most from this proximity, as the study findings show that overcomplexity is frequent among seniors [37] and that regimen complexities are higher in older adults with worse socio-economic and health conditions [31]. Additionally, insufficient pharmacotherapy understanding was high, especially among older adults with low levels of education and dependency on medication use [41]. These findings reinforce the need for pharmacists’ intervention regarding older peoples’ medication.

### 4.4. Pharmacists’ Role

Only four [12,34,40,42] of the 17 included studies mention pharmacists’ intervention: two studies focused directly on regimen complexity simplification while the other focused on MHRPS, with MRC being one of the variables. Elliot [34] demonstrated that a clinical pharmacist’s simplification of older inpatients’ medication is feasible when previous training about simplification is provided. In addition, Elliot et al. [12] concluded that after an educational intervention, a pharmacist-led medication review reduced the impact of hospitalisation on the complexity of older patients’ medication regimens. Furthermore, Moczygemba et al. [40] obtained results that show that a telephone MTM telephone program from a pharmacist reduced MHRPs. Finally, Pouranayatihosseinabad et al. [42] concluded that pharmacists could use the MRCI to identify older adults with complex medication regimens, but they failed to show significant benefits of RMMRs in reducing MRC. However, other of the included studies refer to pharmacists’ potential role in MRC: Elliot et al. [8] concluded that “most regimens had potential to be simplified by a clinical pharmacist review”; Lakey et al. [36] mentioned that “Educational strategies are needed to increase awareness of the pharmacist’s role in facilitating medication management and the option of simplifying complex regimens” (p. 1011); Lindquist et al. [37] stated that *“*health care professionals need to be aware of how patients are taking their medications. … another option would be to partner with pharmacists in reducing medication regimen complexity*”* (p.96); and Linnebur et al. [38] indicated that “our results suggest a need for pharmacist review of the patient’s entire medication regimen … to assess and reduce complexity to a manageable level for the patient if possible” (p. 1545).

One aspect that has to be mentioned under this topic is that the included studies were performed in many countries, where factors like national policies and culture may influence the recognition of pharmacists as a trusted profession in the community and for other health care providers, and therefore may be responsible for the differences observed in pharmacists’ roles [54]. This fact may explain why Australia and the USA were the most representative countries in this review, with three of the four studies mentioning pharmacists’ intervention being performed in Australia [12,34,42].

Given all the above, the present review highlights that the pharmacist’s active role in improving MRC in the older population has been minimal. Nevertheless, the little evidence where pharmacists had an active role showed that medication regimen simplifications are feasible and emphasise the pharmacist’s role to achieve them. However, it is also essential to bear in mind that several studies point out that previous educational sessions for pharmacists are necessary to raise awareness of this topic and give them the skills and practice to minimise regimen complexities [12,34,36]. Even so, a vital opponent to achieve regimen simplification in daily practice seems to be the lack of time of healthcare professionals [12,34]. Having this in mind, pharmacists can, however, take regimen simplifications into account in a more general way whenever they perform OTC advice in their daily routine, and more carefully when performing medication review services.

At last, one of the biggest challenges seems to be multidisciplinary collaboration. Among the findings of the studies, non-acceptance of recommendations by the prescribers is mentioned as one of the most common reasons for noncompliance with suggested regimen changes. As difficulties in the relationship between pharmacists/physicians are well known, we think it is also imperative to sensitise physicians to this subject and make clear that the ultimate goal of this collaboration is health gain, including optimising patients’ health care.

Based on the present review findings, it seems that, until now, pharmacists have not played a relevant role in older people’s MRC. For this reason, future high-quality research should focus on this subject, and in particular should include community pharmacists’ interventions and the resulting possible benefits, not only for patients, e.g., in terms of safety, clinical outcomes and quality of life, but also for the healthcare system, in particular in terms of cost reductions.

### 4.5. Limitations

Although the search was conducted in three major databases, it is always possible that some studies have not been included. Scanning the reference list of the included studies only added one study to the selected ones, indicating that selection bias was minimal. The keywords and synonyms used in the search strategy may have been too restrictive, which may have led to the possible loss of some papers. Despite language bias being frequently reported, only one study has been rejected based on language in our review. Publication bias may also have occurred because only published full papers were included, leading to possibly missing relevant information. Moreover, the included studies were heterogeneous in study design, setting, data collection method, pharmacists’ role and outcomes, which made the comparison difficult and meta-analysis impossible. Finally, it should be noted that this review took under consideration studies performed until October 2019, that is, in a pre-COVID-19 pandemic period. During the COVID-19 pandemic, pharmacists faced new approaches and had to adapt their routine procedures, therefore their role on older adults’ MRC may have been different, and it may be reviewed and eventually compared to the pre-pandemic period.

## 5. Conclusions

Old age is often synonymous with multiple comorbidities and consequently polypharmacy and complex medication regimens. As the latter has been associated with several negative outcomes, particularly in the older population, an effort should be made to reduce MRC whenever possible. Pharmacists may play a relevant role at this point after previous training, which has, however, been underexplored. There is almost no research on pharmacists’ intervention on older people’s MRC; that which does exist is of moderate quality. This aspect leaves an open door for future high-quality evidence investigations on pharmacists’ interventions and their relation to better outcomes. Therefore, pharmacists should be provided with the necessary skills, either during graduation or in post-graduate education and training programs, and encouraged to assess the possibility of simplifying the medication regimen in their daily routine or even on a service-based remuneration model.

## Figures and Tables

**Figure 1 ijerph-18-08824-f001:**
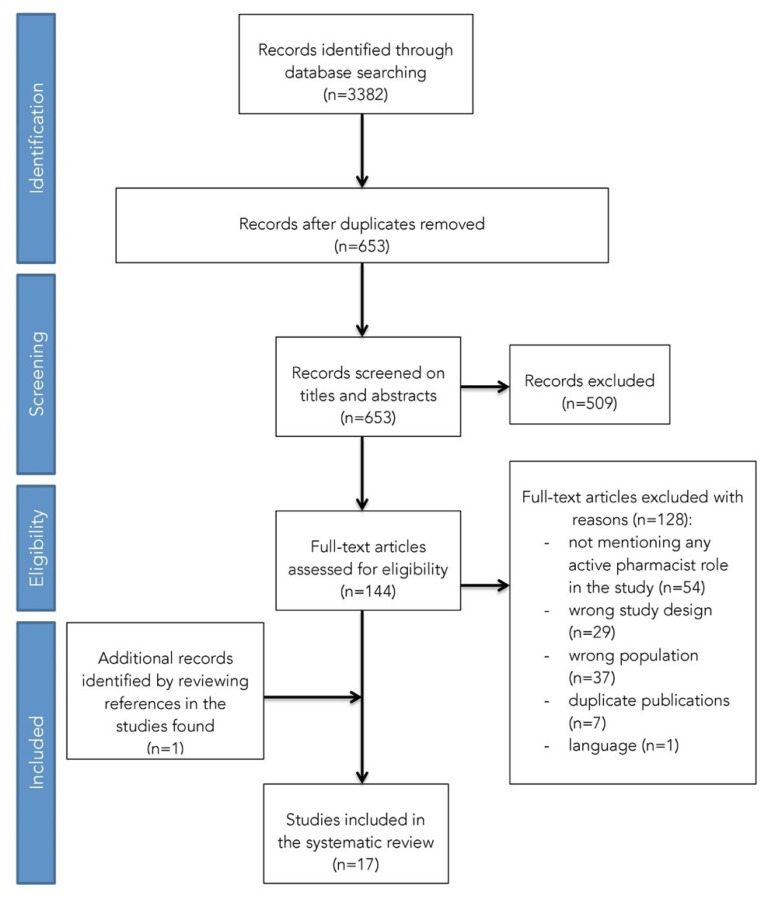
Study selection process using the PRISMA flowchart.

**Table 1 ijerph-18-08824-t001:** Description of the included studies: country, study design, participants’ demographics, setting and study aim.

Author, Year	Country	Study Design	Participants’ Demographics			Setting	Study Aim
			No. of Participants (n)	Mean Age, Years (SD) [Range]	Female		
Acurcio et al. [31], 2009	Brazil	Epidemiological cross-sectional	377	72.4 (61–102)	69.2%	Home visits.	To examine factors associated with therapeutic regimen complexity of drug prescriptions for elderly people.
Bazargan et al. [32], 2017	USA	Cross-sectional study	400	73.5 (7) (65–94)	65%	Sixteen predominantly African-American churches in SPA6 of Los Angeles County.	To examine the association between adherence to drug regimens and a spectrum of medication-related factors, including polypharmacy, medication regimen complexity, use of PIMs, knowledge about their therapeutic purpose and instructions of proper medication use.
Chang et al. [33], 2017	Australia	Retrospective cohort study	100	82 (9.15)	60%	General medical units of a tertiary care hospital.	To assess the changes in the MRCI before and after hospitalisation. To examine the prevalence of prescribing PIMs at the time of hospital discharge, using the 2015 Beers Criteria.
Elliot [34], 2012	Australia	Cohort study	205	81.3 (8.0)	58%	Two acute general medicine wards and two subacute aged care wards at a major metropolitan public hospital.	To explore the feasibility of incorporating medication regimen simplifications into routine clinical pharmacists’ care for older hospital inpatients, and to identify barriers to regimen simplification.
Elliot et al. [10], 2011	Australia	Cohort study	186 Acute wards: 115 Subacute wards: 71	Acute wards: 79 (77–80) Subacute wards: 81 (80–83)	Acute wards: 59% Subacute wards: 55%	Two acute general medicine wards and two subacute aged care wards at a large public hospital.	To investigate the impact of hospitalisation on the complexity of older patients’ medication regimens, and to determine whether discharge medication regimens could be simplified.
Elliot et al. [12], 2013	Australia	Cohort study	391 Pre-intervention: 186 Intervention: 205	Pre-intervention: 79.7 (8.2) Intervention: 81.3 (8.0)	57.8% Pre-intervention: 57.5% Intervention: 58.%	Two acute general medicine wards and two subacute aged care wards at a major metropolitan public hospital.	To investigate the impact of pharmacists’ medication reviews, together with an educational intervention targeting inpatient clinical pharmacists and junior medical officers on the increase in medication regimen complexity during hospitalisation.
Kroenke et al. [35], 1990	USA	Prospective controlled trial	79 Intervention: 38 Control: 41	Intervention: 72.3 Control: 71.4	40.5% Intervention: 39.5% Control: 41.5%	Internal Medicine Clinic at Brooke Army Medical Center.	To determine the effectiveness of specific feedback to prescribing physicians in reducing polypharmacy in elderly outpatients.
Lakey et al. [36], 2009	USA	Cross-sectional	109	85.9 (5.1) (73–98)	79.8%	Continuing care retirement community in Seattle.	To assess older adults’ current use of, knowledge of and preferences for medication management tools and supports.
Lindquist et al. [37], 2014	USA	Cross-sectional	200	79.6 (6.4) (70–100)	58%	Home visits after discharge from Northwestern Memorial Hospital.	To determine whether seniors consolidate their home medications or if there is evidence of unnecessary regimen complexity.
Linnebur et al. [38], 2014	USA	Retrospective cross-sectional	200 CA: 100 CO: 100	CA: 74.3 (7.4) CO: 79.7 (6.1)	78.5% CA: 76% CO: 81%	Ambulatory clinics at the University of CA and the University of CO.	To evaluate the entire medication regimen of older adults with depression, and determine potential targets to simplify the regimen and improve adherence.
Mansur et al. [39], 2012	Israel	Cohort study	212	81.1 (7.3) (66–103)	61.8%	Acute Geriatric Ward at the Beilinson Hospital, Rabin Medical Center.	To test the convergent, discriminant and predictive validity of the MRCI in older hospitalised patients with varying functional and cognitive levels.
Moczygemba et al. [40], 2012	USA	Quasi-experimental	120 Intervention: 60 Control: 60	Intervention: 71.2 (7.5) Control: 73.9 (8.0)	60% Intervention: 48.3% Control: 71.7%	Telephone consultation.	To determine the impact of telephone MTM on MHRPs, medication adherence and total drug costs for Medicare Part D participants.
Pinto et al. [41], 2016	Brazil	Cross-sectional	227	71.4	70.9%	Two PHUs in the municipality of Belo Horizonte.	To evaluate the level of understanding of pharmacotherapy and the associated factors amongst older people in two PHUs.
Pouranayatihosseinabad et al. [42], 2018	Australia	Retrospective observational study	285	85.5 (7.7)	68%	Residential ACFs.	To investigate the impact of RMMRs on simplifying medication regimen complexity in Australian ACF residents using the MRCI.
Sevilla-Sánchez et al. [43], 2017	Spain	Prospective cross-sectional study	235	86.80 (5.37)	65.50%	AGU in a second-level hospital.	To determine the prevalence of PIMs among patients with advanced chronic conditions and palliative care needs, and to analyse the associated risk factors and resulting clinical consequences.
Wimmer et al. [44], 2014	Australia	Prospective cohort	163 Readmitted: 99 Not readmitted: 64	Readmitted: 84.9 (6.2) Not readmitted: 85.6 (6.74)	72.4% Readmitted: 68.7% Not readmitted: 78.1%	GEM unit of a public hospital in Adelaide.	To investigate the association between discharge medication regimen complexity and unplanned re-hospitalisation over 12 months.
Wimmer et al. [22], 2014	Australia	Prospective cohort	163 DD home: 87 DD NCS: 76	85.2 (6.4) (71–101) DD home: 84.6 (6.9) DD NCS: 85.8 (5.8)	72.4% DD home: 68.7% DD NCS: 77.6%	GEM unit at the Queen Elizabeth Hospital.	To investigate the association between polypharmacy and medication regimen complexity with hospital discharge destination among older people.

ACFs, aged care facilities; AGU, acute care geriatric unit; CA, California San Diego; CO, Colorado Anschutz Medical Campus; DD, discharge destination; GEM, geriatric evaluation and management; MHRP, medication- and health-related problems; MRCI, medication regimen complexity index; MTM, medication therapy management; NCS, non-community setting; PHUs, primary health care units; RMMRs, residential medication management reviews; and SPA6, Service Planning Area 6.

**Table 2 ijerph-18-08824-t002:** Description of the included studies: medication data, pharmacists’ role and main outcomes.

Author, Year	Medication Data			Pharmacists’ Role in MRC	Main Outcomes
	Source	Type of Medication Included	Instrument to Assess MRC		
Acurcio et al. [31], 2009	Medical prescription	Prescription medication	MCI	Household data collection, after previous training.	MCI ranged from 1 to 24 (mean = 6.1; median = 5.0). Nr. of drugs prescribed (> 2), less schooling, worse perception of health and lower benefit payment associated positively with greater complexity (*p* < 0.05). An association was observed between RC and failure to use some drugs in the preceding 15 days (*p* = 0.34).
Bazargan et al. [32], 2017	The brown bag method	Prescription medication	MRCI	Evaluation of any medication duplication; application of the Beers Criteria to document the number of PIM use; and comparison of subjects perceived purpose of each prescription drug with all therapeutic indications used in clinical practice.	The mean value of the MRCI was 15.1 (SD = 8.2; minimum = 2.5; and maximum = 55.5). Of the participants, 70% (278) engaged in PIM use and used at least one medication that was classified as “Avoid” (27%) and “Use Conditionally” (43%) through the Beers Criteria. Participants with increased knowledge about the therapeutic purpose of the dosage regimen were almost seven times more likely to adhere to their medication.
Chang et al. [33], 2017	Hospital electronic medical record system	Prescription medication	MRCI	Medication history record.	The mean MRCI increased from 28.70 at the time of admission to 32.46 at discharge (*p* = 0.007). Hospitalisation resulted in a statistically significant reduction in the prevalence of the use of PIMs.
Elliot [34], 2012	Hospital medical record	Long-term, short-term and “when required” medication	Not mentioned	Pharmacists were encouraged to minimise RC and discuss simplifications with hospital doctors and patients. Afterwards, they were asked to indicate if they reviewed the patient’s RC (and why not) and whether regimen changes were considered or attempted. Changes had to be recorded and whether they were successfully implemented (and if not why).	Pharmacists reviewed medication RC for 173/205 (84.4%) patients and identified 149 potential changes to reduce RC for 79/173 (45.7%) reviewed patients. Ninety-four (63.1%) changes were successfully implemented in 54/205 (26.3%) patients.
Elliot et al. [10], 2011	Pre-admission: medication history on admission within the patient’s hospital medical record Discharge: discharge prescription	Long-term, short-term and “when required” medication	MRCI	Review of discharge medication regimens and identification of any potential change that could make the regimen simpler.	MRCI scores increased by 22% (18 to 22; *p* < 0.0001) for regularly scheduled long-term medications and 32% (21 to 27; *p* < 0.001) for all medications. Ninety simplifications to regularly long-term medications were proposed and 84 (93%) were rated as feasible and likely to have the same or similar outcome.
Elliot et al. [12], 2013	Pre-admission: hospital medical record Discharge: discharge prescription	Long-term, short-term and “when required” medication	MRCI	Clinical pharmacists were encouraged to review RC and make recommendations to hospital medical officers to simplify medication regimens when clinically appropriate.	MRCI score for all medications: pre-intervention patients = 20.7 (SD = 12.5); intervention patients = 21.7 (SD = 11.6). MRCI score for all regularly scheduled long-term medication: pre-intervention patients = 18.2 (SD = 11.2); intervention patients = 19.1 (SD = 10.3). The mean increase in MRCI score between admission and discharge was significantly smaller in the intervention patients than in the usual care patients (2.5 vs. 4.0; *p* = 0.02, adjusted difference 1.6; 95% CI 0.3, 2.9).
Kroenke et al. [35], 1990	Patients were asked to bring all prescription bottles to the interview, where medications were used regularly and the dosage schedules were confirmed	Prescription medication	A complexity score was calculated by summing the different dosage intervals, weighted for frequency	A clinical pharmacist interviewed patients to determine the precise drug regimen. Investigating physicians discussed and agreed upon recommendations that might reduce polypharmacy.	All four indices of polypharmacy improved in the intervention group (7.2 vs. 6.6; *p* = 0.007). Physicians complied 100% with recommendations to simplify a dosage schedule, 62% to substitute a new drug for the old one and only 40% with recommendations to stop a medication (*p* = 0.04).
Lakey et al. [36], 2009	Participants were asked to show containers for all medication taken in the week before	Prescription medication, non-prescription medication and herbal products	The frequency of dosing for each medication was summed to calculate a complexity score	The investigating pharmacist performed data collection.	Medication complexity score: only for prescription drugs = 5.0 ± 3.8; total = 8.3 ± 4.4.
Lindquist et al. [37], 2014	Subjects were asked to show how they take their medication on a typical day and all medications were registered and compared to discharge instructions	Medication subjects take a day	The number of times medications were taken in a 24 h period for each subject was calculated	A pharmacist acted as one of two coders, determining the fewest number of times a day that a patient could take the regimen.	Home medication regimens could be simplified for 85 (42.5%) subjects.
Linnebur et al. [38], 2014	Electronic health record	Depression medications, other prescription medications and over-the-counter (OTC) medications	MRCI/patient-level MRCI (pMRCI)	A clinical pharmacist coded the pMRCI using an electronic data capture tool that calculated three subscores.	Individual pMRCI scores average: 17.62 (CA) and 19.36 (CO). Dosing frequency contributed to 57–58% of the MRCI score, with patients facing an average of 7–8 unique dosing frequencies in their regimen.
Mansur et al. [39], 2012	Patient interview on admission. Retrospectively, patients’ medical files were reviewed to calculate their MRCI score for discharge medication regimens	Long-term discharge medication	MRCI	The pharmacist interviewed the patients to collect clinical and demographical data as well as patients’ medication on admission.	Mean (SD) MRCI at discharge: 30.27 (13.95). The MRCI score was strongly correlated with the number of medications (r = 0.94; *p* < 0.001) and the number of daily doses (r = 0.87; *p* < 0.001), and increased as the number of medications taken ≥ 3 times a day increased (27.35; 34.45 and 43.00 for none, 1 and 2 drugs; *p* < 0.001).
Moczygemba et al. [40], 2012	Electronic medical records and prescription claims	Prescription medication	MRCI	Pharmacists reviewed the patient’s electronic medical records to identify potential MHRPs. Furtherly, in the intervention group, an MTM telephone consultation was made and recommendations were given to the patient to resolve MHRPs.	MRCI intervention group: 21.5 ± 7.8. MRCI control group: 22.8 ± 6.9.
Pinto et al. [41], 2016	Prescribed medication that each individual had in his hand at the time of the interview	Prescription medication	MRCI	Resident pharmacists and academics studying pharmacy were part of the team carrying out data collection.	Mean value of the rate of complexity: 22.7 (DP = 10.9; CV = 48.0%), with a minimum of 4.0 and a maximum of 65.5.
Pouranayatihosseinabad et al. [42], 2018	RMMR report	Prescription medication	MRCI	Accredited pharmacists compiled RMMR reports, which included pharmacists’ findings and recommendations.	The median MRCI at baseline was 25.5 (19.0–32.5). The main contributing factor to the MRCI score was dosing frequency. Pharmacists made 764 recommendations, of which GPs accepted 74.5%. There were no significant differences in the MRCI scores after pharmacists’ recommendations (*p* = 0.53) or after GPs’ acceptance of these recommendations (*p* = 0.07) compared to baseline.
Sevilla-Sánchez et al. [43], 2017	Not mentioned	Routine chronic medication	MRCI	A multidisciplinary team consisting of a pharmacist and two physicians determined the PIMs.	MRCI (mean; SD) = 38.00 (16.54). Of the population, 88.50% had at least one STOPP criterion and 97.40% had some criterion according to the MAI criteria. The identified risk factors for the existence of PIMs were: insomnia, anxiety–depressive disorders, falls, pain, excessive polypharmacy and therapeutic complexity.
Wimmer et al. [44], 2014	Hospital separation summary recorded in the OACIS	All prescription and non-prescription medications, nutritional supplements, health products, dermatological preparations and short-term medications were considered	MRCI	The same pharmacist researcher extracted all medication data.	Mean MRCI for: - discharged patients: 27.86 (SD = 11.63) - readmitted: 28.01 (SD = 12.48) - not readmitted: 27.62 (SD = 10.26). The MRCI was not significantly different in patients who were readmitted and not readmitted over 12 months (mean difference = −0.39; 95% CI = −4.09 to 3.30).
Wimmer et al. [22], 2014	Hospital separation summary recorded in OACIS	Prescription, non-prescription and CAMs	MRCI	The same pharmacist researcher extracted all medication data.	Patients discharged directly to home: mean MRCI = 26.1; SD 9.7. Patients discharged to NCSs: mean MRCI = 29.9; SD 9.7. High medication RC (MRCI > 35) inversely associated with discharge directly to home (RR 0.39; 95% CI 0.20–0.73). Polypharmacy (≥ 9 medications) not significantly associated with discharge directly to home (RR 0.97; 95% CI 0.53–1.58).

CAMs, complementary and alternative medications; GP, general practitioner; MCI, medication complexity index; MRCI, medication regimen complexity index; NCS, non-community setting; OACIS, Open Architecture Clinical Information System; PIMs, potentially inappropriate medications; RC, regimen complexity; and RMMR, residential medication management review.

## Data Availability

Not applicable.

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
