# Peer review of "Pharmacists’ Role in Older Adults’ Medication Regimen Complexity: A Systematic Review"

_ijerph, 2021, doi:10.3390/ijerph18168824_

Round 1

Reviewer 1 Report

This study is well carried out and has followed the PRISMA guidelines. The Introduction is adequate although it should refer to other indices such as Defined Daily Dosage and Drug Burden Index, which are different but related indices and should indicate they have been excluded.

The literature search has been comprehensively performed and the results fully presented. The Discussion is sound. The Conclusion however should also contain what the authors conclude from their review as well as generally related matters.

The English grammar needs some attention throughout the manuscript largely based on selecting more appropriate words or tenses. For example in the Conclusion "synonym" should be "synonymous with"  In the next sentence "last" should be "latter",  

Author Response

This study is well carried out and has followed the PRISMA guidelines.

The Introduction is adequate although it should refer to other indices such as Defined Daily Dosage and Drug Burden Index, which are different but related indices and should indicate they have been excluded.

Author’s answer: Firstly, the authors would like to thank you for your comments and recommendations. The authors are aware of the existence of the referred indices; however, it is important to mention that they were not intentionally excluded but no reference was found relating such indices with medication regimen complexity, nor in a previous literature search nor in the further systematic review. A possible explanation may be the fact that, on one hand, the Defined Daily Dosage is a theoretical index that does not reflect the prescribed daily dose and is not established for all medication, and on the other hand, that the Drug Burden Index is a measure of exposure to specific drugs (anticholinergic and sedative medications), related to medication burden. Although medication complexity has, so far, no clear definition and its evaluation may be based on polypharmacy, like medication burden, it seems to us to be talking about two different concepts [1,2]. Thus, in this context, the authors are willing not to add any reference to these indices in the “Introduction” of this manuscript, because it diverges from the main topic.

  1. Gnijidic, D.; Tinetti, M.; Allore, H.G. Assessing medication burden and polypharmacy: finding the perfect measure. Expert Rev Clin Pharmacol. 2017; 10(4): 345-347. doi: 10.1080/17512433.2017.1301206
  2. Boye, K.S.; Mody, R.; Lage, M.J.; Douglas, S.; Pate, H. Chronic medication burden and complexity for US patients with Type 2 Diabetes Treated with Glucose-Lowering Agents. Diabetes Ther. 2020; 11(7): 1513-1525. doi: 10.1007/s13300-020-00838-6

The literature search has been comprehensively performed and the results fully presented. The Discussion is sound.

The Conclusion however should also contain what the authors conclude from their review as well as generally related matters.

Author’s answer: The authors acknowledged that the conclusion required improvement and it was revised, following your comment (please see text, lines 358 and 362-365).

The English grammar needs some attention throughout the manuscript largely based on selecting more appropriate words or tenses. For example in the Conclusion "synonym" should be "synonymous with"  In the next sentence "last" should be "latter", 

Author’s answer: The manuscript was revised by a native English speaker and the English grammar improved, including the suggested word changes. The certificate of review will be attached.

Reviewer 2 Report

The manuscript is thoroughly describing the role of pharmacists in medication regimen complexity. The methods and description of the results are sound.

I will ask authors to provide a translational outlook of the information provided in the manuscript.

Author Response

The manuscript is thoroughly describing the role of pharmacists in medication regimen complexity. The methods and description of the results are sound.

I will ask authors to provide a translational outlook of the information provided in the manuscript.

Author’s answer: Firstly, the authors would like to thank you for your comments and recommendations. In accordance with the suggestion made be Reviewer #1, a translational outlook was added to the conclusion (please see text, lines 358 and 362-365).

Reviewer 3 Report

The paper reviewed the role of pharmacists playing on the older population’s MRC. The topic follows the journal. The author has lots of studies in related fields and is qualified for the review of this direction. 
Some points
There are too many small paragraphs in the first part, please re-plan according to the theme of each paragraph.
Line 61. As of October 2019, it is not the latest. Please update to the latest. Or during the pandemic, the role of the pharmacists will be different, please also point out and review.
Many countries are listed in Table 1. I think that the policies, culture, wealth gap, communication tools etc. of different countries may have a great impact on the Pharmacists’ role. Whether the author mentions this and compares it in the discussion. 
Line 109. “previously”
Line 110. “afterward”
Line 220 “the heterogeneity”
Line 226 “an equal number”
Line 232 “especially”
Line 261 “take advantage of” 
Line 326 “possibly”
Line 331 “a synonym”

Author Response

The paper reviewed the role of pharmacists playing on the older population’s MRC. The topic follows the journal. The author has lots of studies in related fields and is qualified for the review of this direction. 

Some points

There are too many small paragraphs in the first part, please re-plan according to the theme of each paragraph.

Author’s answer: Firstly, the authors would like to thank you for your comments and recommendations. The introduction was reviewed to avoid small paragraphs, which has only three paragraphs now.

Line 61. As of October 2019, it is not the latest. Please update to the latest. Or during the pandemic, the role of the pharmacists will be different, please also point out and review.

Author’s answer: In agreement, the referred recommendation was added to the limitations section (please see text, lines 345-350).

Many countries are listed in Table 1. I think that the policies, culture, wealth gap, communication tools etc. of different countries may have a great impact on the Pharmacists’ role. Whether the author mentions this and compares it in the discussion.

Author’s answer: This information was added to the discussion section, as recommended (please see text, lines 301-307, reference 54).

Line 109. “previously”
Line 110. “afterward”
Line 220 “the heterogeneity”
Line 226 “an equal number”
Line 232 “especially”
Line 261 “take advantage of” 
Line 326 “possibly”
Line 331 “a synonym

Author’s answer: The terms were changed in the manuscript, in agreement (please see text, lines 112 (former 109), 113 (former 110), 224 (former 220), 230 (former 226), 237 (former 232), 267 (former 261), 342 (former 326)). The last word (line 353 - former 331) was changed to synonymous, in accordance with the suggestion made by Reviewer#1 and the native English speaker who revised the manuscript.

Reviewer 4 Report

The methods used and their limitations were clearly presented. The discussion allows us to make concrete proposals on possible directions for future research. However, the descriptions of the characteristics of the articles in this review do not highlight the data collection methods (quantitative or qualitative methods) in all the articles. These methods influence the quality of the results (interviews with pharmacists, other healthcare professionals, patients, or data collection from existing databases, etc.). At least this could have been noted in the limitations and enriched recommendations for future research in the discussion.

Author Response

The methods used and their limitations were clearly presented. The discussion allows us to make concrete proposals on possible directions for future research.

However, the descriptions of the characteristics of the articles in this review do not highlight the data collection methods (quantitative or qualitative methods) in all the articles. These methods influence the quality of the results (interviews with pharmacists, other healthcare professionals, patients, or data collection from existing databases, etc.). At least this could have been noted in the limitations and enriched recommendations for future research in the discussion.

Author’s answer: Firstly, the authors would like to thank you for your comments and recommendations. In agreement, this information was included in the limitations section (please see text, line 343).
